# White LED Lighting Increases the Root Productivity of *Panax ginseng* C. A. Meyer in a Hydroponic Cultivation System of a Plant Factory

**DOI:** 10.3390/biology12081052

**Published:** 2023-07-26

**Authors:** Se-Hee Kim, Jae-Hoon Park, Eui-Joo Kim, Jung-Min Lee, Ji-Won Park, Yoon-Seo Kim, Gyu-Ri Kim, Ju-Seon Lee, Eung-Pill Lee, Young-Han You

**Affiliations:** 1Department of Biological Science, Kongju National University, Gongju 32588, Republic of Korea; ksh41631@smail.kongju.ac.kr (S.-H.K.); kn5314@smail.kongju.ac.kr (J.-H.P.); euijoo@kongju.ac.kr (E.-J.K.); ljm@smail.kongju.ac.kr (J.-M.L.); ecopark@kongju.ac.kr (J.-W.P.); 201502761@smail.kongju.ac.kr (Y.-S.K.); gyuri128@smail.kongju.ac.kr (G.-R.K.); wntjs2433@smail.kongju.ac.kr (J.-S.L.); 2National Ecosystem Survey Team, National Institute of Ecology, Seocheon 33657, Republic of Korea; ecolee21@nie.re.kr

**Keywords:** plant factory, medicinal crop, light spectra, multivariate analysis, intercellular CO_2_ partial pressure, root biomass, hydroponic cultivation

## Abstract

**Simple Summary:**

Today, *Panax ginseng* C. A. Meyer, which has high economic value, is cultivated for the purpose of using the shoot as well as the root, which is better known as the medicinal properties. However, when ginseng is grown outdoors, the quantity and quality of the crop are negatively affected by the climatic environment. In this study, we compared and analyzed the physiological and growth responses of *P. ginseng* under different LED spectra in a plant factory to achieve continuous and increased productivity. Red and yellow light effectively increased shoot biomass, whereas white light effectively increased root biomass. Furthermore, the intercellular CO_2_ partial pressure was identified as the most significant physiological variable contributing to root production. Research on light spectra in controlled environments can provide insights into increasing *P. ginseng* production and contribute to the understanding of the physiological and growth responses of shade-tolerant plants such as *P. ginseng*.

**Abstract:**

To identify effective light spectra for increasing the productivity of *Panax ginseng*, we conducted experiments in a controlled environment using a hydroponic cultivation system in a plant factory. We investigated the effect of single LEDs (red, blue, and yellow) and mixed LEDs (red + blue and red + blue + white). The relationships between four light spectra (red, blue, yellow, and white) and physiological responses (net photosynthetic rate, stomata conductance, transpiration rate, and intercellular CO_2_ partial pressure), as well as growth responses (shoot and root biomass), were analyzed using multivariate statistical analysis. Among the four physiological response variables, shoot biomass was not increased by any pathway, and root biomass was increased only by the intercellular CO_2_ partial pressure. Red and yellow light increased shoot biomass, whereas white light promoted an increase in the net photosynthetic rate and enhanced root biomass. In contrast, blue light was less effective than the other light spectra in increasing both shoot and root biomass. Therefore, red and yellow light are the most effective light spectra for increasing shoot biomass and white light is effective for increasing root biomass in a plant factory that uses artificial LED lighting. Furthermore, the intercellular CO_2_ partial pressure is an important physiological variable for increasing the root biomass of *P. ginseng*.

## 1. Introduction

Light is one of the most important environmental factors affecting plant growth and development. Studying the effects of light on plants is important because plants use different wavelengths of light for various physiological processes such as photosynthesis and photomorphogenesis. Different wavelengths of light can trigger various physiological responses in plants by activating specific light receptors, such as phytochromes, cryptochromes, and phototropins [1,2]. Although red light is essential for photosynthesis, blue light plays a significant role in photomorphogenesis, including stem elongation and leaf expansion [3,4]. Green and yellow light are less efficient but can affect plant growth and development [5,6,7]. Understanding how different wavelengths of light influence plant physiology enables the provision of optimal light environments for various plant species.

Because the optimal spectral combination, light intensity, photoperiod, and dark period differ among plant species, a controlled light environment is required to understand the physiological responses of plants to light spectra. Plant factories can provide such an environment. These indoor facilities grow crops under controlled environmental conditions, including light spectra, light intensity, temperature, humidity, etc. [8]. Unlike field cultivation, which is constrained by light environmental conditions and requires large amounts of water and fertilizer, plant factories offer several advantages for studying the effects of light spectra [8,9]. In plant factories, precise control over the spectral bands, intensity, and photoperiod provides an optimal environment for plant growth and development, enabling the consistent production of high-quality crops [10]. Additionally, it reduces the damage caused by pests and diseases, thereby minimizing the need for chemical inputs, such as pesticides and other chemicals [10,11]. While plant factories can offer many benefits, such as year-round cultivation, high yields, and consistent quality, issues regarding the energy required to maintain systems such as lighting, heating, and cooling remain [8]. Some studies have found that energy consumption in plant factories can be much higher than that in traditional greenhouses because of the energy required for artificial lighting [12,13]. Therefore, from an energy efficiency perspective, cultivating shade-tolerant plant species in plant factories can offer potential advantages for sustainable agriculture [14,15].

*Panax ginseng* C. A. Meyer is a shade-tolerant species located in the lower part of the forest that grows well under low light intensity and filtered sunlight [16] and is suitable for cultivation in plant factories [14]. In addition, *P. ginseng* is a representative medicinal crop in the Orient that has been used in traditional medicine for centuries due to its various pharmacological effects, such as improving cognitive function, reducing stress, and strengthening the immune system for human health [17]. In the past, *P. ginseng* roots were cultivated for medicinal purposes; however, recently, studies on using the aboveground parts, such as leaves and shoot, have been conducted [18,19], and the consumption of *P. ginseng* has increased worldwide [20,21]. However, since the traditional cultivation of *Panax* genus plants is mainly done in open fields, the quality and yield may be affected depending on the climatic environment [22]. Therefore, plants such as *P. ginseng*, which are vulnerable to climate change, can be grown stably through constant environmental controls in plant factories [10].

In this study, the effect of the LED light spectra of the plant factory on the physiology and growth response of *P. ginseng* was quantitatively investigated using multivariate analysis. In this study, we aimed to identify the physiological variables involved in the increase in shoot and root biomass of *P. ginseng*. Furthermore, we propose light spectra that are effective in increasing the production of shoots and roots of *P. ginseng* with high economic value.

## 2. Materials and Methods

### 2.1. Planting and Cultivation of P. ginseng

The *P. ginseng* used in the experiment were one-year-old seedlings purchased from Baeknyeon Ginseng Agriculture Co., Ltd. (Geumsan-eup, Republic of Korea) in March 2022. *P. ginseng* with no visible damage, similar sizes and weights, and no sprouting buds were selected. *P. ginseng* was transplanted onto a plate measuring 30 cm (W) × 20 cm (L) × 0.5 cm (H), with a uniform distance of 4 cm between individuals. During this process, the buds of the *P. ginseng* plant protruded above the plate. The planting plate was placed in a plastic tank with dimensions of 32 cm (W) × 22 cm (L) × 15 cm (H) made of an opaque material and positioned inside the cultivation chamber. The cultivation chamber used was 1.2 m (W) × 0.6 m (L) × 0.7 m (H), and divided into three layers. For each layer, an LED light source (1.2 m (W) × 0.6 m (L) × 0.52 m (H)) from Parus Co. (Cheonan-si, Republic of Korea) was installed on the ceiling of the chamber. The cultivation period was from April to May 2022 for a total of eight weeks.

To compare the physiological and growth responses of *P. ginseng* under different light conditions, we used the deep flow technique (DFT), a hydroponic cultivation method that enables the control of nutrients and oxygen (Figure 1). The nutrient solution was prepared by diluting the hydroponic cultivation nutrient solution (Cosil Comprehensive Herbicide, Co., Seoul, Republic of Korea) 1/100 in tap water that had been left to stabilize for one week and was supplied once every two weeks. The pH of the nutrient solution was 6.37 ± 0.11, and the EC was 1.04 ± 0.34 dS/m. Oxygen was continuously supplied by installing air pumps and air stone inside the tank, maintaining dissolved oxygen (DO) levels within the range of 11.69 ± 0.33 mg/L, which were checked once a day. The pH and DO were measured using a multiparameter water quality sonde (YSI 600XL probe), and the EC was measured using a multi-range conductivity meter (HI-9033). The temperature and humidity within the smart farm were maintained at 20.9 ± 2.6 °C and 54.9 ± 9.4%, respectively, using a cooling fan and humidifier. Temperature and humidity were monitored every 30 min using an environmental measurement system (LCSEMS; Parus Co., Daejeon, Republic of Korea).

### 2.2. Light Environment Conditions

We used LED lights (Parus Co.) with two types of mixed light (red + blue (RB) or red + blue + white (RBW)) and three types of single light (red (R), blue (B), and yellow (Y)) (Figure 2, Table 1). RB and RBW lights, which have been proven to be effective light sources and are commonly used in indoor cultivation research, were used for comparison with single light sources. Single red, blue, and yellow lights were chosen to identify the physiological and growth responses that were maximized in *P. ginseng* under each light condition.

The spectra and intensities of each light source were measured using a spectrometer (LI- 180A; LI-COR, Lincoln, NE, USA) (Figure 2, Table 2). RB light (400–500 nm and 630–700 nm) displayed peaks at 450 nm and 660 nm, RBW light (400–700 nm) exhibited peaks at 450 nm and 660 nm and showed a uniform distribution of green (500–560 nm) and yellow (560–630 nm) wavelengths owing to white light. Red light (630–700 nm) showed a peak at 660 nm, blue light (400–500 nm) at 450 nm, and yellow light (560–630 nm) at 595 nm. Additionally, all light sources used in this study did not include far-red light (Table 2). Light intensity (PPFD) inside the chamber was measured using a quantum sensor (LI-250A; LI-COR) (Table 2). Using R, B, and Y together with RB and RBW, which are mainly used for crop cultivation in plant factories, we attempted to determine the effect of each wavelength on the physiological response of *P. ginseng*. The light intensity for RB, RBW, R, and B was maintained at about 100 μmol m^−2^ s^−1^, which is known to be a suitable condition for *P. ginseng* growth [23,24,25]. In contrast, Y, which is known to be of low importance for the growth of green plants in the visible light region, was maintained at approximately 20–30 μmol m^−2^ s^−1^, similar to the amount of light in the understory layer where *P. ginseng* lives [26]. The photoperiod during the cultivation period was set at 16 h d^−1^.

### 2.3. Measurement Parameters

Measurements were taken in the sixth week of cultivation using a photosynthesis measurement device with a broad leaf chamber (LCi Ultra Compact Photosynthesis System, ADC Bio Scientific Ltd., Hoddesdon, UK) to examine changes in the physiological responses related to photosynthesis in *P. ginseng*. The net photosynthetic rate (Pn; μmol m^−2^ s^−1^), transpiration rate (E; mmol m^−2^ s^−1^), intercellular CO_2_ partial pressure (Ci; vpm), and stomatal conductance (gs; mol m^−2^ s^−1^) were measured (Table 1). Additionally, *P. ginseng* was harvested after the cultivation period—wilted individuals were excluded from the measurement—to compare the morphological responses of *P. ginseng* to the light source. Biomass represents the total quantity or weight of living organisms per unit area, indicating plant productivity, and is usually measured as dry weight [27,28]. The final productivity of *P. ginseng* was assessed using shoot dry weight (SDW; g) and root dry weight (RDW; g) (Table 1). The shoots and roots of *P. ginseng* were separated, dried at 80 °C in a drying oven for 48 h, and then weighed. In addition, the S/R ratio was calculated to understand the investment ratio of the shoot and root according to the light spectra (Table 1).

### 2.4. Data Analysis

To investigate the differences in the physiological and growth responses of *P. ginseng* according to the light spectra, we performed the Kolmogorov–Smirnov test to confirm the normality of the data. Because the data did not have a normal distribution, we conducted a nonparametric statistical analysis using the Kruskal–Wallis test. Furthermore, path analysis was used to examine the causal relationships between light spectra and the physiological and growth responses of *P. ginseng*. The variables included four light spectra (R, B, Y, and W), four physiological response variables (Pn, E, Ci, and gs), and two growth response variables (SDW and RDW) (Table 2). For data organization, the light spectral distribution variables were transformed into binary variables (1 = “presence of the specific light spectra,” 0 = “absence of the specific light spectra”), and the physiological and growth response variable data were arranged according to the light spectra. All variables were standardized to ensure comparability with each other. Path analysis was performed using the proposed model (Figure 3). The model included the effect of light spectra on physiological response, the effect between physiological variables, the effect of physiological response on growth, and the effect of light spectra on growth (Figure 3). The physiological responses showed a causal relationship from Pn, gs to E, Ci, which led to their division into primary physiological responses (Pn, gs) and secondary physiological responses (E, Ci). The model was evaluated using the Chi-squared test (2–3) and goodness-of-fit indices (comparative fit index (CFI); 0.9–1, normed fit index (NFI); 0.9–1, Bollen’s incremental fit index (IFI); 0.9–1, relative noncentrality index (RNI); 0.9–1, and goodness of fit index (GFI); 0.9–1) and all showed good fit (Appendix A) [29,30,31]. Based on the results of the path analysis, only significant paths among the effects of the independent variables on the dependent variables were displayed (*p* < 0.05).

Statistica sofrware (version 7) was used for the Kruskal–Wallis test, and JASP software (version 0.16.4) was used for path analysis.

## 3. Results

### 3.1. Physiological Responses

The Pn of *P. ginseng* was 1.5 (B) to 1.9 (R) times higher in the RBW light with W added than in the RB, R, B, and Y lights (Figure 4a). In particular, the fact that the Pn of RBW light with added W was higher than that of RB light with two light sources suggests that W (containing green and yellow light) makes a significant contribution to photosynthesis. The gs of *P. ginseng* was 2.5 (R) to 4.7 (RB) times higher under Y light than under RB, R, and B light (Figure 4b), and the E was 2.8 (RBW) to 5.4 (RB) times higher under Y light than under RB and B light (Figure 4c). In addition, the gs and E exhibited similar trends for all light spectra, suggesting that gs has a direct effect on E. In addition, the Ci showed similar levels under R, B, and Y lights, and Ci under RB light was 1.6 (R and Y) to 1.8 (B) higher than that under R, B, and Y lights (Figure 4d).

### 3.2. Growth Responses

The SDW of *P. ginseng* was 1.2 (RBW) to 1.8 (B) times heavier than that of other light treatments in Y light, and tended to increase more in RBW light with W added than in RB, R, and B light (Figure 5a). In addition, the RDW of *P. ginseng* was 1.1 (RB) to 1.9 (B) times heavier than the other light treatments in the RBW light, and it tended to increase in RBW light with W added compared to RB light (Figure 5b). In addition, *P. ginseng* invested more in the shoot than in the root in B and Y light, and invested more in the root in RB, RBW, and R light (Figure 5c).

### 3.3. Pathways Affecting Physiological and Growth Responses

R and B did not affect the primary physiological responses (Pn, gs) but did affect the secondary physiological responses (E, Ci) (Figure 6). R and B decreased the E and increased the Ci (R→E, B→E, R→Ci, B→Ci). Y increased the gs, E, and Ci (Y→gs, Y→E, Y→Ci) but did not affect the Pn. W increased the Pn and E, decreased the Ci (W→Pn, W→E, W→Ci), but did not affect the gs. In addition, gs increased the E (gs→E) and had no effect on the Ci (Figure 6), and Pn decreased the E and Ci (Pn→E, Pn→Ci). In addition, the Pn, gs, and E did not directly affect the SDW and RDW; only Ci increased the RDW of *P. ginseng* (Ci→RDW).

R and Y increased the SDW (R→ADW, Y→ADW; Figure 6). In contrast, B and W affected the root of *P. ginseng* more than the shoot. B decreased and W increased the RDW (B→RDW, W→RDW).

## 4. Discussion

### 4.1. The Role of Light Spectra in Physiological Responses

The physiological response of *P. ginseng* according to light spectra was revealed through path analysis (Figure 6). Among the four light spectra, only W increased the Pn of *P. ginseng* (Figure 6). In particular, the increased Pn in RBW compared to RB proved that supplementing W is effective in improving the Pn (Figure 4). However, white light alone can decrease the net photosynthetic rate of lettuce, a heliophyte plant, compared to RB [32], indicating that W may be less effective in increasing the Pn of plants. Therefore, based on our results, it can be inferred that Pn increased because RB was supplemented with W.

Increased gs signifies a greater opening of stomata, which enhances the exchange of H_2_O, CO_2_, and other gases between the interior and exterior of the leaves [33,34]. In our results, the gs did not differ among the RB, RBW, R, and B, but increased only under Y (Figure 4). Additionally, Y increased stomatal conductance, leading to an increase in transpiration rate (Figure 6). These results suggest that, compared to the general stomatal aperture induced by other light spectra, Y promoted a greater opening of stomata, resulting in an elevated E (Figure 6). In contrast, in the case of *Nicotiana tabacum*, a heliophyte plant, the gs was the highest under red light among various light sources, which differs from our findings [35]. This difference may be due to the physiological responses arising from the adaptations of the shade-tolerant plant *P. ginseng* and the heliophyte plant *N. tabacum* to their respective habitats [36].

Furthermore, all light spectra affected the Ci (Figure 6). R, B, and Y increased the Ci, whereas white light decreased it (Figure 6). The decrease in the Ci in response to W can be attributed to the pathway where an increase in the Pn leads to a decrease in the Ci (Figure 6). These findings suggest that white light facilitates carbon assimilation, leading to the consumption of CO_2_ [37]. Conversely, the Pn increases were not observed under R, B, or Y, indicating relatively low carbon fixation (Figure 6). As a result, R, B and Y had less CO_2_ fixation than white light, so there must have been more CO_2_ that was not fixed inside the leaf, and it is considered that this led to the emergence of a pathway that increased Ci.

### 4.2. The Role of Light Spectra in Growth Responses

Plant biomass is considered one of the parameters indicating growth and yield [27,38]. In the path analysis conducted based on the observation results in a controlled environment, the effect of light spectra on the growth of *P. ginseng* was shown through a direct path (Figure 6).

The SDW of *P. ginseng* exhibited an increasing trend under RB, RBW, R, and Y compared with that under blue light (Figure 5). Moreover, path analysis revealed that, unlike B and W, R and Y increased shoot biomass (Figure 6). This aligns with the findings that R results in the highest shoot biomass in *Rubus hongnoensis* [39], and Y increases stem biomass in chrysanthemums and tomatoes [40], leading to enhanced shoot growth. It indicated that yellow or red light can effectively increase the aboveground parts biomass. Moreover, our results on *P. ginseng* provided important insights into its potential for Y utilization. *P. ginseng* naturally inhabits the understory of forests and receives limited light, with a high proportion of green and yellow (500–600 nm) wavelengths [41,42]. In such a limited light environment, plants undergo morphological changes to maximize the absorption of red and blue light essential for photosynthesis [34], and they invest more in shoot growth than in root growth to maximize light absorption [43,44]. Therefore, the increased shoot biomass of *P. ginseng* under Y in the present study can be interpreted as an adaptation to shaded environments (Figure 6). Additionally, despite the yellow light having approximately 1/3 lower light intensity compared to other light treatments, *P. ginseng* showed similar shoot and root biomass under Y (Figure 5). This finding further supports the notion of *P. ginseng*’s growth response being adapted to a shaded environment. In contrast, heliophyte plants such as *Solanum lycopersicum* and *Arabidopsis thaliana* did not exhibit increased shoot or leaf biomass under higher-light-intensity yellow light such as was observed in this study [36,45]. This suggests that the adaptation of heliophytes to yellow light may be less effective compared to shade-tolerant plants like *P. ginseng*. Furthermore, these results suggest that, when shade tolerance crops such as *P. ginseng* are cultivated in the future, the supplementation of yellow light should be considered in addition to red light, and they show the possibility of the efficient use of yellow light.

In contrast, the RDW of *P. ginseng* showed an increasing trend under RB, RBW, R, and Y compared with that under B (Figure 5). Also, B decreased the RDW of *P. ginseng* (Figure 6), indicating that the increase in RDW was less under B (high proportion of blue light) compared to other lights (Figure 5). These results suggest that, although a higher proportion of blue light increases the number of stolons and improves root production in the heliophyte plant *Epilobium hirsutum* L. [46], it can hinder root biomass in shade-tolerant plants such as *P. ginseng*. In contrast, white light increased the root biomass of *P. ginseng*, which was attributed to pathways involving the Pn and Ci (Figure 6). These results indicated that white light increased the photosynthetic rate of *P. ginseng*, promoting carbon fixation reactions (Figure 6) and resulting in the accumulation of more photosynthetic products in non-photosynthetic organs, the roots, leading to an increase in the RDW [34,47]. Furthermore, the four physiological responses (Pn, gs, and E) had no significant effect on the increase in the SDW and RDW, and only Ci contributed to an increase in the RDW (Figure 6). These results indicate that Ci is an important physiological variable for increasing the RDW of *P. ginseng*.

## 5. Conclusions

This study confirmed that various light spectra influenced the physiological and growth responses of *P. ginseng* through 18 direct and indirect pathways. These four light spectra have diverse effects on the physiological and growth responses of *P. ginseng*. The supplementation of RB with W increased the Pn. Y most effectively increased the gs. In contrast, R and B had a minimal impact on the Pn and gs. All light spectra affected the Ci, which was identified as a key factor influencing RDW increase in *P. ginseng*. R and Y had a greater impact on the increase in the SDW, indicating the importance of R and Y in the light adaptation of shade-tolerant plants such as *P. ginseng*. Additionally, W enhanced carbon fixation and increased the RDW, whereas B hindered the increase in RDW when applied in high proportions.

Therefore, to enhance the shoot production of *P. ginseng*, it is necessary to increase the ratio of R rather than B or consider the supply of Y. Moreover, for plants in which root production is crucial, the selection of light sources that include W in the RB should be considered. Furthermore, research on light spectra in controlled environments is expected to contribute to the understanding of the physiological responses and adaptations in shade-tolerant plants.

## Figures and Tables

**Figure 1 biology-12-01052-f001:**
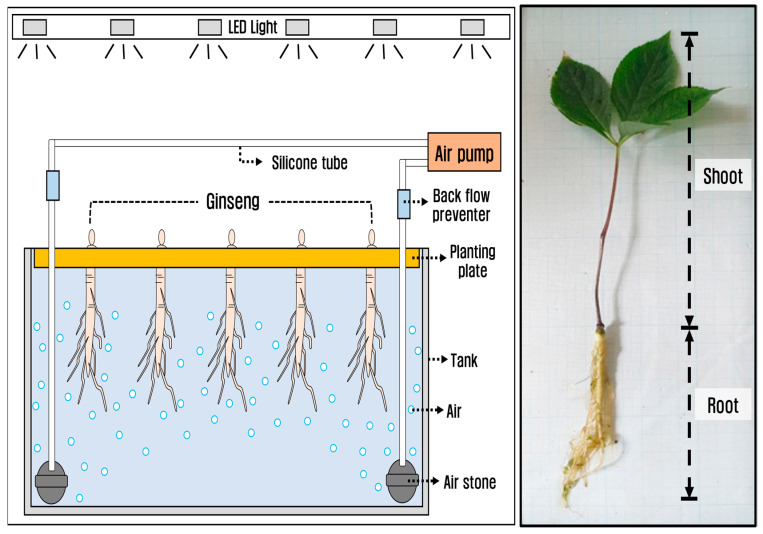
Schematic diagram of hydroponic cultivation of *Panax ginseng*.

**Figure 2 biology-12-01052-f002:**
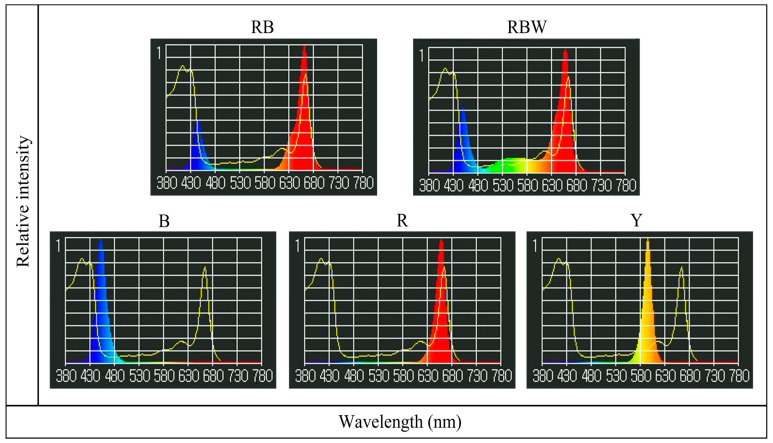
The spectral distributions of LED light sources. The graph shows the distribution and peak of wavelengths based on the total light intensity of each LED. RB: Red + Blue light; RBW: Red + Blue + White light; B: Blue light; R: Red light; Y: Yellow light. The yellow line indicates the absorption spectrum of chlorophyll a as a reference spectrum (the spectral images used the data provided by Li-180A).

**Figure 3 biology-12-01052-f003:**
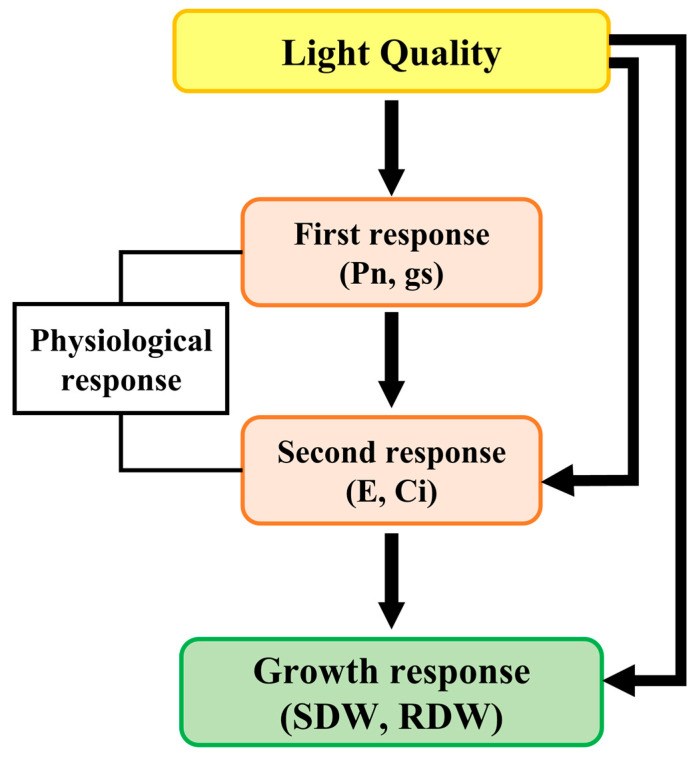
Path analysis model of physiological and growth responses of hydroponic cultivation Panax ginseng according to light spectra. The starting point of the arrow indicates independent variables, and the endpoint indicates dependent variables. The solid line indicates the direct effect.

**Figure 4 biology-12-01052-f004:**
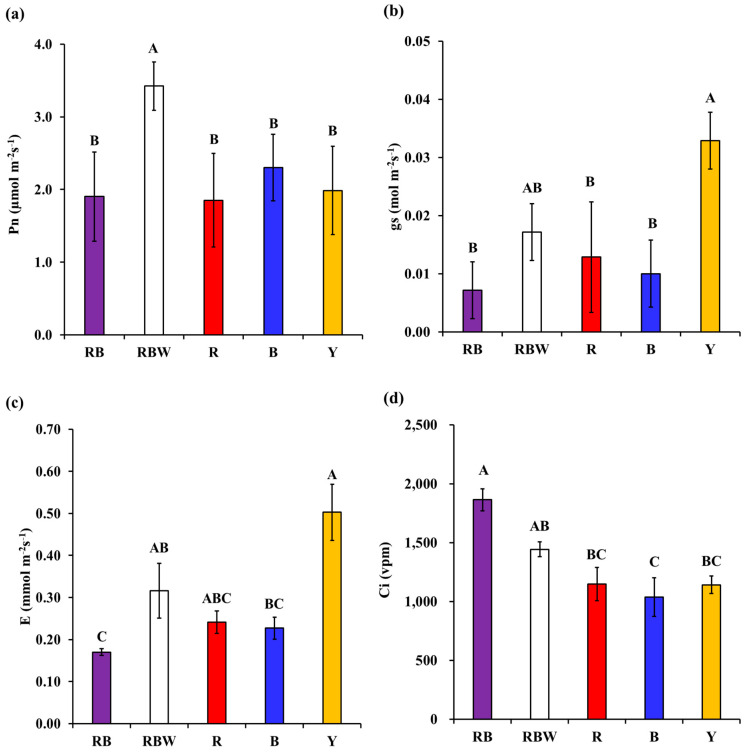
Physiological responses of hydroponic cultivation *Panax ginseng* according to light spectra. Pn: net photosynthetic rate (**a**); gs: stomatal conductance (**b**); E: transpiration rate (**c**); Ci: intercelluar CO_2_ partial pressure (**d**). Letters on the bars indicate significant difference among light spectra (*p* < 0.05).

**Figure 5 biology-12-01052-f005:**
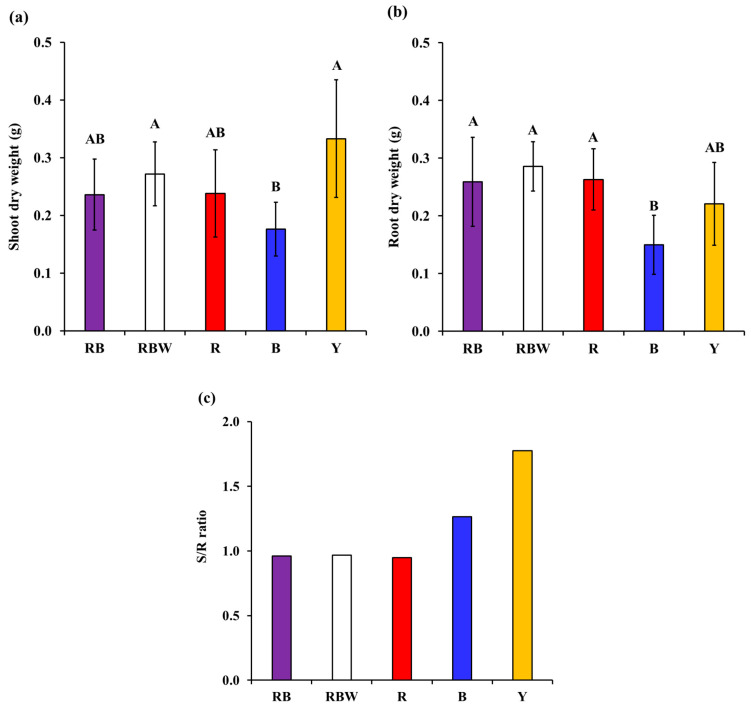
Shoot and root growth responses of hydroponic cultivation *Panax ginseng* according to light spectra. Shoot dry weight (**a**), root dry weight (**b**), S/R ratio (**c**). Letters on the bars indicate significant difference among light spectra (*p* < 0.05).

**Figure 6 biology-12-01052-f006:**
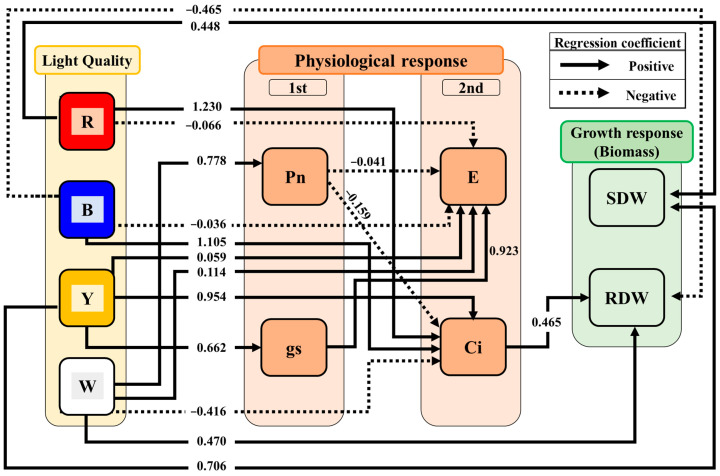
Results from the proposed causal relationship between physiological and growth responses to light spectra in hydroponic cultivation *Panax ginseng*. All lines have significant *p*-values (*p* < 0.05). Solid arrows indicate positive paths and dotted arrows indicate negative paths. Each regression relationship was considered as a direct path, and the continuation of regression relationships was regarded as an indirect path. Numbers on each line indicate standardized regression coefficients. The model fit was significant (X^2^ = 23.686, df = 11, *p* = 0.014).

**Table 1 biology-12-01052-t001:** Abbreviations of physiological and growth response factors.

Abbreviation	Description	Units
R	Red light	μmol m^−2^ s^−1^
B	Blue light	μmol m^−2^ s^−1^
Y	Yellow light	μmol m^−2^ s^−1^
W	White light	μmol m^−2^ s^−1^
Pn	Net photosynthetic rate	μmol m^−2^ s^−1^
E	Transpiration rate	μmol m^−2^ s^−1^
Ci	Intercellular CO_2_ partial pressure	vpm
gs	Stomatal conductance	mol m-m^−2^ s^−1^
SDW	Shoot dry weight	g
RDW	Root dry weight	g
S/R ratio	Shoot dry weight to root dry weight ratio	-

**Table 2 biology-12-01052-t002:** Spectral ratio and light intensity (PPFD) for single and mixed LEDs.

Light Source	Spectral Ratio (%)				PPFD (μmol m^−2^ s^−1^)
	Blue (400–500 nm)	Green (500–560 nm)	Yellow (560–630 nm)	Red (630–700 nm)	
RB	29.9	0.4	4.1	65.6	98.8 ± 9.3
RBW	19.7	9.5	16.6	54.2	115.3 ± 12.6
B	98.5	1	0.2	0.3	112.8 ± 22.4
Y	0.5	1.2	97.4	0.9	27.7 ± 4.0
R	0.3	0.2	4.8	94.7	114.4 ± 20.0

## Data Availability

The data presented in this study are available in the Appendix A.

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
