# Peer review of "White LED Lighting Increases the Root Productivity of Panax ginseng C. A. Meyer in a Hydroponic Cultivation System of a Plant Factory"

_biology, 2023, doi:10.3390/biology12081052_

Round 1
Reviewer 1 Report
The manuscript entitled “White LED light increase the growth of Panax ginseng C. A. Meyer root productivity in a plant factory” by Se-Hee Kim et al. describes the effect of plant illumination with monochromatic light and its combinations on the development of the root system of Panax ginseng grown in hydroponics system.
The studied LED light wavebands include, red, blue, yellow, red + blue and white light combinations. Light effects were evaluated by several parameters including, net photosynthetic rate (Pn), stomatal transduction (gs), transpiration rate (E), intercelular CO2 pressure (ci), shoot dry wight (SDW) and root dry weight (RDW).
Title of the manuscript is interesting as beneficial light effects are usually reserved for specific light colors and not for LED white light. Plant factory is also mentioned in the title although it is a bit misleading as work was actually done in hydroponics.
Upon first reading manuscript appears to be a nice flowing and interesting text, well composed and written in fine English language, with proper explanation of light an their sources and well done statistics supporting data presentation, provided in appealing illustrations. In few places there are some hints of troubles which authors neutralized and the only remark was the use of capital letters to mark statistical significance in illustrations which are of the same level as marks on the X axis. Second inspection revealed some week points which need to be corrected prior to acceptance.
1. It seems that authors are hiding the fact that experiments were done in a hydroponics, mentioned only in material and methods. I would expect it to be mentioned in the title, abstract, and among the keywords. It is needed to be added to the title otherwise it would be misleading readers. There is one more question in connection with the utilized hydroponics, is the root system in the dark or some of the applied light perhaps partly illuminates the roots.
2. Authors should find a place and proper form to inform readers that far red light was not studied. It was not used as an individual light and its presence in the emission specter of employed white light sources is apparently minimal. Panax ginseng as a species adjusted to
shade is expected to be sensitive to far red /red ratio changes, amply demonstrated as important in other plant species. Authors can at least declare at the end of text their intention to investigate red light interactions in their future studies. Far red light sources are now widely available.
3. The choice of employed physiological parameters is not good. They are not of the same physiological importance and their evaluation as if they were equal is wrong. Valuable and important parameters here are Pm, SDW and RDW, showing the physiological productivity of plants and the shoot vs. root partitioning of assimilate. In my opinion parameters gs and E are of no great importance in a hydroponic system and parameter ci is anyhow already connected with the Pn parameter as stated by authors.
Therefore some statements and conclusions in the manuscript based on investigated parameters are a bit far fetched
4. Statistic seems to be well done and in more than expected details. However the simple fact that data interpretation required Kolmogorov-Smirnof non parametric testing indicates some problems either with the experimental design and protocols or less likely with plant responses.
Results would be more trustworthy if authors could provide data on production/ accumulation of a root metabolite, a compound specific for the root system. But that is a different story. Authors have a good experimental approach and I believe they will continue their studies employing other parameters in their future work.
After making some minor adjustments, manuscript is acceptable for publication.
Reviewer
Reviewer 2 Report
Broad comments:
- --- Kim et al. examined the effect of monochromatic and broad spectrum light on the growth of ginseng. The objective of the present manuscript is significant for the future LED-based plant cultivation however, the experimental design and treatments have to be improved for consideration for publication of this manuscript.
-
L109: please list the ingredients of the hydroponic solution with pH and EC.
L111: please provide info on DO. Any air flow in this chamber?
L146: The authors explained why the using low intensity of Y than others, however, this made data interpretation difficult and the author barely highlight the impact of Y on photosynthetic and growth responses. Nearly 1/3 of Y light energy led to comparable growth response to other spectrum.
Relevant literatures:
https://doi.org/10.3390/plants12132457
https://doi.org/10.1371/journal.pone.0247380
L154: what were the light intensities of each light treatments used in the photosynthetic measurement?
L156: what was the chamber used with the photosynthetic measurement device?
---The Introduction in this manuscript could benefit from better flow and readability as it currently presents some challenges for readers to follow smoothly.
